# Porcine Astrocytes and Their Relevance for Translational Neurotrauma Research

**DOI:** 10.3390/biomedicines11092388

**Published:** 2023-08-26

**Authors:** Erin M. Purvis, Natalia Fedorczak, Annette Prah, Daniel Han, John C. O’Donnell

**Affiliations:** 1Center for Neurotrauma, Neurodegeneration & Restoration, Corporal Michael J. Crescenz Veterans Affairs Medical Center, Philadelphia, PA 19104, USAdaniel.han@pennmedicine.upenn.edu (D.H.); 2Center for Brain Injury & Repair, Department of Neurosurgery, Perelman School of Medicine, University of Pennsylvania, Philadelphia, PA 19104, USA; 3Department of Neuroscience, Perelman School of Medicine, University of Pennsylvania, Philadelphia, PA 19104, USA

**Keywords:** swine, pig, porcine, astrocytes, glia, translational neurotrauma

## Abstract

Astrocytes are essential to virtually all brain processes, from ion homeostasis to neurovascular coupling to metabolism, and even play an active role in signaling and plasticity. Astrocytic dysfunction can be devastating to neighboring neurons made inherently vulnerable by their polarized, excitable membranes. Therefore, correcting astrocyte dysfunction is an attractive therapeutic target to enhance neuroprotection and recovery following acquired brain injury. However, the translation of such therapeutic strategies is hindered by a knowledge base dependent almost entirely on rodent data. To facilitate additional astrocytic research in the translatable pig model, we present a review of astrocyte findings from pig studies of health and disease. We hope that this review can serve as a road map for intrepid pig researchers interested in studying astrocyte biology.

## 1. Introduction

Astrocytes, the most abundant cell type in the brain, were once believed to play only a supporting role in brain function. However, research over the last few decades has revealed that they are active participants in nearly all facets of brain activity. They maintain ion gradient homeostasis essential for brain function, prevent edema, remove glutamate from the extracellular space, couple neuronal activity to changes in blood flow and glucose uptake, and engage directly in signaling events and plasticity mechanisms. As the hub between the vasculature and the synapse, they are also central to anabolic and catabolic metabolism in the brain, processing glucose for energy substrates (glycogen, lactate, etc.) or to synthesize essential molecules like glutamate, as well as providing antioxidant protection to neighboring neurons and performing the essential task of fixing NH_4_ to allow incorporation of nitrogen into biological molecules. Neurotrauma and neurodegenerative diseases that impact these essential astrocytic functions can be devastating to the brain and the organism as a whole (for reviews, see [1,2,3,4,5,6,7,8]). Their central role in facilitating both healthy brain function and the mechanisms of neurotrauma and neurodegenerative disease, combined with their resilience relative to neighboring neurons, make astrocytes a very attractive therapeutic target [9,10,11]. Translating such therapies will require investigations in a large animal model like pigs to investigate the mechanisms and manifestations of human injuries more closely.

The overwhelming majority of our astrocyte knowledge stems from research in rodents due to their low cost and ease-of-use. However, recent genetic brain atlas comparisons between mice, pigs, and humans revealed greater variability between humans and mice relative to humans and pigs [12], and the distribution of splice variants of glutamate transporters (a signature family of astrocytic proteins) also shows a greater similarity between pigs and humans relative to mice and humans [13], suggesting that translation of knowledge even at the level of cell biology should involve a large animal model like the pig at some stage. In addition to the cellular similarities between humans and pigs (relative to rodents), pigs are high-fidelity models for the translational study of acquired brain injury due to their large gyrencephalic brains, structural similarities in limbic, subcortical, diencephalic, and brainstem regions, high white-to-gray matter ratio, and similar basal cistern geometry, among other features [14,15,16,17,18,19,20,21,22]. Due to their relatively large brain mass, pigs offer the unique opportunity to recreate rotational acceleration injury, the predominant mechanism of human traumatic brain injury (TBI), which in turn presents the unique opportunity to study TBI manifestations that cannot be recreated in rodents, such as coma and other disorders of consciousness [14,23]. Head rotational acceleration TBI in pigs also recreates acute crises such as apnea and increased intracranial pressure, which, coupled with their large size providing compatibility with clinical neuromonitoring equipment, makes them an ideal model for studying neurocritical care [24].

Beyond direct brain injury, we will also discuss studies of astrocytes in pigs that have experienced indirect brain injuries through mechanisms such as cardiac arrest, sepsis, and others. While we will have inevitably overlooked some research works, we believe that this neurotrauma-forward review provides a near-comprehensive assessment of studies that have assessed astrocytes in pigs. Such a review would be impossible in rodent literature, further emphasizing the relative lack of, and need for, astrocyte research in pigs.

## 2. Investigations of Astrocyte Structure and Function in the Porcine Brain

To date, there have been several studies that have investigated the structure and function of astrocytes in the pig brain. Porcine astrocytes have been investigated in vivo, in slice culture, histologically, and using cell culture techniques. Here, we review the details of these studies to highlight what is known about porcine astrocytes. We only discuss elements of these publications that are relevant to astrocytes in the porcine brain. Furthermore, as it can be difficult to find histological antibodies for pig specimens, we summarize all available antibody information from the reviewed papers in Appendix A.

### 2.1. Glutamate Transporters

The astrocytic sodium-dependent glutamate transporters GLAST (GLutamate ASpartate Transporter) and GLT-1 (GLutamate Transporter-1) are responsible for clearing glutamate from the extracellular space. Glutamate is the primary excitatory neurotransmitter in the mammalian brain and also, paradoxically, a potent neurotoxin. Therefore, one of the most obvious and important tasks astrocytic glutamate transporters perform is the constant clearance of glutamate from the extracellular space to prevent excitotoxic neuronal death. However, astrocytic glutamate transporters and associated calcium signals appear integral to many brain functions. Glutamate uptake into fine astrocytic processes results in local reversal of the sodium/calcium exchanger that positions astrocytic mitochondria near GLT-1 clusters that are servicing active synapses [25]. Aside from facilitating glutamate oxidation, local calcium signals in fine astrocytic processes appear between neighboring mitochondria, very rarely extending past a mitochondrion. However, when astrocytic mitochondria were damaged via transient oxygen/glucose deprivation in organotypic hippocampal slices (rat), most of these calcium signaling events extended beyond neighboring mitochondria [26]. These observations suggest that glutamate transporters may provide additional information integration via astrocytic calcium signaling between local synapses facilitated by mitochondria in fine astrocytic processes and that trauma can disrupt this communication. Astrocytic glutamate transporters are vitally important to brain function, but little is known about them outside of rodent models.

The Pow lab published a detailed article describing the expression patterns of the glutamate transporters GLAST, GLT-1alpha, and GLT-1v in the Large White/Landrace porcine brain [13]. They conducted immunohistochemistry for these three transporter proteins as well as GFAP in the coronal and sagittal sections of the porcine brain. They reported the highest level of immunoreactivity to GLAST in the cerebellum, with strong labeling in the Bergmann glial cells of the molecular layer and less labeling of astrocytes in the granular cell layer. The porcine hippocampus also expressed high levels of GLAST, with significant labeling in areas CA1, CA2, CA3, and the dentate regions. Additionally, there was GLAST labeling in the molecular layers of the fascia dentate and the hilar region of the dentate such that certain layers stained more strongly for GLAST than others. GLAST labeling was evident in all cortical gray matter layers. The motor cortex was evenly labeled for GLAST expression, while the frontal and temporal cortices had patchy labeling with some strongly immunoreactive areas interspersed with areas of weakly labeled tissue. These patchy areas were not contained within any specific cortical regions but rather seemed to be present in all cortical layers. GLAST co-labeled with GFAP in these patchy cortical regions. There was strong labeling of GLT-1alpha in the cortex, putamen, hippocampus, thalamus, and cerebellum. There was little labeling of GLT-1alpha in the forebrain white matter (Figure 1). In the cerebellum, GLT-1alpha was strongly labeled in Bergmann glial cells of the molecular layer, in astrocytes in the granular layer, and in the deep cerebellar nuclei. There was also significant GLT-1alpha labeling in hippocampal areas CA1, CA2, CA3, and the dentate gyrus, with the strongest dentate labeling occurring in the molecular layer. Immunolabeling of GLT-1alpha was strong but patchy in cortical areas such as the frontal cortex and temporal cortex. They also reported immunolabeling of the porcine brain for GLT-1v, but this labeling was seen in oligodendrocytes rather than astrocytes. GLT1-alpha expression was restricted to gray matter regions, whereas GLT-1v expression was restricted to white matter regions. They also reported that GLAST and GLT-1alpha expression sometimes, but not always, co-localized with GFAP staining. Overall, these results indicate that, as has been observed in other mammals, different brain regions have different glutamate transport properties and that GLAST and GLT-1apha glutamate transporters sometimes co-localized with GFAP in the porcine brain.

The Pow lab followed up on this with a publication describing the associated expression of the glutamate transporter GLAST and the astrocyte intermediate filament protein GFAP in the 1-day-old piglet brain [27]. In this study, they demonstrated co-immunoprecipitation of GFAP and GLAST in the cortex and cerebellum of the piglet brain. They immunoprecipitated total piglet brain lysate with anti-GFAP antibodies and detected GLAST, and immunoprecipitated total piglet brain lysate with anti-GLAST antibodies and detected GFAP. These results suggested an in vivo interaction between GFAP and GLAST in the piglet brain. They also performed immunohistochemistry to reveal that GFAP was strongly expressed by astrocytes in the gray and white matter of the piglet brain. GLAST was highly expressed in the cortical gray matter, hippocampus, thalamus, and hypothalamus. High magnification imaging revealed a close spatial association of these two markers, with GFAP expression in the core or cytoskeleton of the cell and GLAST expression in the plasma membrane. This publication also included data on changes in GFAP and GLAST expression following hypoxia in the piglet brain, which is discussed in the hypoxia–ischemia injury section below. These results, along with results gathered from experiments on the rat brain, led the authors to hypothesize that GFAP stabilizes astrocyte processes, which could help to anchor GLAST in the plasma membrane.

This group also demonstrated that the GLAST1c splice variant is expressed in the piglet brain [28]. RT-PCR and Western blot were used to identify the expression of GLAST1c in the pig cortex. They also used immunohistochemistry to demonstrate the presence of GLAST1c in Bergmann glial cells of the molecular layer, astrocytes of the granule cell layer, and cells in the white matter layer of the pig cerebellum.

### 2.2. GFAP (Glial Fibrillary Acidic Protein)

Throughout this review, it will become evident that histological staining of GFAP is by far the most common technique used to study astrocytes, primarily via morphological analyses or simply by measuring stain intensity. This section describes studies in pigs that focused on GFAP itself.

Blechinberg and colleagues published a report detailing the expression of the GFAP isoforms GFAPα, GFAPε, and GFAPκ in the adult and developing porcine brain [29]. They utilized real-time PCR (RT-PCR) to analyze and quantify the mRNA expression of these three GFAP isoforms in adult pig cortical tissue. They report that in the adult pig cortex, the expression of GFAPα is approximately 100-fold higher than the expression of GFAPκ and roughly 60-fold higher than the expression of GFAPε. The expression of GFAPε is almost double the expression of GFAPκ in the adult pig cortex. They also performed RT-PCR analysis of these three GFAP isoforms in fetal brain tissue comprised of the hippocampus, cortex, basal ganglia, cerebellum, and brain stem of embryonic day 40, 60, 80, 100, and 115 fetal pigs. They reported that mRNA levels of GFAPα, GFAPε, and GFAPκ increase across porcine development. The highest GFAP expression was observed in the brainstem and cerebellum brain regions, and the lowest GFAP expression was in the cortex. They reported the highest increase in GFAP expression occurred between e100 and e115 in the brainstem. The mRNA ratio of GFAPα/GFAPε is consistent across development, with GFAPα around 100-fold higher than GFAPε. The mRNA ratio of GFAPα/GFAPκ in the basal ganglia and brainstem increases on e115 compared to e60, whereas the GFAPα/GFAPκ ratio in the cerebellum increases between e60 and e80 and then decreases again. The GFAPκ/GFAPε mRNA ratio decreases from e60 to e115. Overall, the GFAPα/GFAPκ and GFAPε/GFAPκ ratios are higher in the adult cortex compared with the brain during fetal pig development. The results of this study indicate that GFAP is first expressed around e40 and is tightly regulated during fetal development. The authors speculate that changing mRNA ratios of the various GFAP isoforms throughout development could reflect cues for glial cell differentiation. Another group employing transcriptomic analyses to study glial differentiation in pigs reported that marker genes of astrocytes, including GFAP and AQP4, are differentially expressed in the male and female porcine brain during development between gestational days 45 and 90 [30].

### 2.3. Inflammatory Signaling

The Busija lab published two reports detailing prostaglandin synthesis in cultured porcine cerebral astrocytes [31,32]. The first report revealed that PGF_2α_ is the predominant prostaglandin produced by porcine astrocytes [31]. These cells were observed to produce minimal amounts of 6-keto-PGF_1α_, PGE_2_, and LTC_4_/D_4_. Application of phorbol 12,13-dibutyrate (PDB), a protein kinase C (PKC) activator, onto astrocyte cultures led to increased levels of PGF_2α_. PDB application did not increase levels of 6-keto-PGF_1α_, PGE_2_, or LTC_4_/D_4_. This PDB-induced increase in PGF_2α_ was prevented when cultures were treated with the drugs indomethacin, quinacrine (phospholipase A_2_/PLA_2_ inhibitor), or isoquinolinylsulfonylmethyl piperazine (PKC inhibitor) at the same time as PDB. Application of 4α-phorbol 12,13-didecanoate (PDD), which does not activate PKC, did not lead to increased levels of PGF_2α_ in porcine astrocyte cultures. These results indicate that the major prostaglandin produced by porcine cerebral astrocytes is PGF_2α_ and that PKC activation increases the production of PGF_2α_ through a mechanism that may involve PLA_2_. In a separate study, they also demonstrated that the administration of interleukin 1α (IL-1α) to cultured porcine astrocytes rapidly increases their production of PGF_2α_ [32]. Using a previously validated protocol, PGF_2α_ was increased two-fold when 11 μg/mL of IL-1α was added and four-fold when 22 μg/mL of IL-1α to astrocyte media. To our knowledge, equivalent dosing in humans has not been determined. Levels of 6-keto-PGF_1α_ and PGE_2_ did not change with IL-1α addition. These results confirm the results of their previous study indicating that PGF_2α_ is the major prostaglandin produced by porcine astrocytes.

The Zimmer lab conducted a study to investigate the cultured porcine astrocyte expression of major histocompatibility complex (MHC) antigens and their ability to induce the proliferation of human T-lymphocytes [33]. They reported that cultured astrocytes harvested from the fetal porcine brain were not autofluorescent, stained negative for CD18, and stained positive for GFAP, CD44, and the NCAM isoform of CD56. Cultured astrocytes also had upregulated levels of MHC class I antigens compared to freshly isolated cells, indicating that the process of culturing cells can lead to upregulated MHC antigens. Cultured astrocytes did not express MHC class II antigens. Cultured astrocytes also induced a proliferative response in human T lymphocytes, which provides relevant information to studies investigating neural xenotransplantation of porcine donor cells that contain an astrocytic population.

Ionescu and colleagues reported that astrocytes harvested from the adult porcine cortex express the markers GFAP, S100β, CD14, and interferon-γ receptor 2 (IFN-γ-R2) [34]. This group also examined the response of astrocytes to human SH-SY5Y neuroblastoma cells when exposed to proinflammatory mediators. When astrocytes were exposed to IFN-γ or a combination of lipopolysaccharide (LPS) and IFN-γ, they were cytotoxic to SH-SY5Y cells (as seen by increased levels of cell death). Exposure to LPS alone did not cause astrocyte toxicity toward SH-SY5Y cells. The authors did not detect increased levels of tumor necrosis factor-α (TNF-α) or nitric oxide (NO) in astrocyte culture media when astrocytes became cytotoxic toward SH-SY5Y cells, indicating that these neurotoxins were not mediating the astrocyte response to SH-SY5Y cells.

The Parfenova group published a detailed report investigating the antioxidant and cytoprotective effects of sulforaphane (SFN) in porcine cortical astrocyte cultures [35]. They demonstrate that adding the pro-inflammatory cytokine TNF-α or excitotoxic glutamate to astrocyte cultures rapidly led to increased production of reactive oxygen species (ROS). Independent application of tiron (a potent superoxide scavenger), apocynin (common Nox inhibitor), DPI (diphenylene iodonium; common Nox inhibitor), GKT137831 (novel Nox4 inhibitor), or SFN blocked ROS elevation in the presence of TNF-α or excessive glutamate. Application of DPI, GKT137831, or SFN to astrocyte cultures also inhibited NADPH oxidase activity under control conditions, TNF-α-induced inflammatory conditions, and excitotoxic glutamate conditions. Application of tiron or SFN to astrocyte cultures reduced DNA fragmentation and cell detachment, two indicators of apoptosis, under TNF-α-induced inflammatory conditions and excitotoxic glutamate conditions. These results indicate that the application of TNF-α or excitotoxic glutamate to astrocyte cultures activates Nox4 NADPH oxidase, which increases ROS, which in turn causes oxidative stress and apoptosis. The inhibitor compounds investigated in this study, including GKT137831 and SFN, exhibit antioxidant and cytoprotective properties toward astrocytes via inhibiting Nox4 activity.

### 2.4. Vascular Regulation

Barnes et al. investigated the presence of the cell surface peptidases aminopeptidase N (AP-N) and dipeptidyl peptidase IV (DPP-IV) in porcine striatal astrocytes [36]. They reported that AP-N, but not DPP-IV, is expressed by astrocytes in the piglet striatum. AP-N was seen in the cell membranes of striatal astrocytic endfeet, including in astrocytes associating with endothelial cells and pericytes. They demonstrated this with immunolabeling and transmission electron microscopy of primary cultures isolated from the postnatal-day-1 piglet striatum and brain sections from the postnatal-day-1 piglet striatum. The presence of AP-N in the cell membranes of striatal astrocytic endfeet suggests that astrocytes may actively participate in the enzymatic processing of peptides in the striatum. Additionally, the localization of AP-N in endothelial cells and pericytes suggests a potential role in the interaction between astrocytes and the neurovascular unit. This may enable striatal astrocytes to regulate the processing and transport of circulating peptide signaling throughout the brain.

Leffler and colleagues investigated the role of astrocytes in the dilation of porcine cerebral arteriole myocytes [36]. In porcine astrocyte cultures, inhibition of the enzyme heme oxygenase (HO) with chromium mesophorphyrin (CrMP) prior to being placed in contact with myocytes caused the elimination of glutamate-induced K_Ca_ channel activation in myocytes. Additionally, treatment of porcine cortical brain slices with the selective astrocyte toxin L-2-α-aminoadipic acid (L-AAA) blocked glutamate-induced dilation of arterioles. These results provide evidence that glutamate-induced dilation of monocyte arterioles is dependent on astrocyte-derived HO.

Authors from this same group then investigated the role of astrocytes in the mediation of Ca^2+^ signaling in arteriolar smooth muscle cells in live brain slices from the porcine cortex [37]. Following astrocyte injury by treating brain slices with L-AAA, glutamate-induced Ca^2+^ spark activation and reduction in intracellular Ca^2+^ concentration in arteriolar smooth muscle cells were both prevented. L-AAA treatment also increased glutamate-stimulated Ca^2+^ wave frequency in smooth muscle cells. This experiment demonstrates that astrocytes modulate glutamate-induced Ca^2+^ sparks, Ca^2+^ waves, and global intracellular Ca^2+^ levels in smooth muscle cells.

This group then investigated the mechanism of glutamate-induced carbon monoxide (CO) production in cultured porcine astrocytes [38]. They demonstrated that administration of glutamate, hemin (an exogenous HO substrate), and ionomycin (Ca^2+^ ionophore that held free intracellular calcium concentration constant) to astrocyte cultures each increased astrocyte CO production in a concentration-dependent manner. Administration of glutamate also increased intracellular calcium concentration in astrocyte cultures in a concentration-dependent manner. When astrocyte cultures were treated with the endoplasmic reticulum Ca^2+^-ATPase blocker thapsigargin, which depleted intracellular Ca^2+^ stores, glutamate-induced Ca^2+^ signaling was blocked, and steady-state intracellular calcium levels were elevated, suggesting that glutamate stimulates Ca^2+^ release from the ER which in turn leads to intracellular calcium concentration. Glutamate did not increase astrocyte CO production when Ca^2+^ levels were held constant with ionomycin, suggesting that glutamate stimulates astrocyte CO production via Ca^2+^. HO was inhibited by CrMP, which blocked glutamate-induced CO production in cultured astrocytes. Additionally, treatment of astrocytes with thapsigargin reduced basal CO levels and blocked glutamate-stimulated CO increase in astrocytes. Treatment of cultures with the calmodulin blocker calmidazolium also blocked glutamate-stimulated CO increase in astrocytes. Overall, these results suggest a mechanism by which glutamate causes the ER to release Ca^2+^, which leads to elevated intracellular Ca^2+^ levels, in turn leading to Ca^2+^-calmodulin-dependent HO activation and CO production in porcine cortical astrocytes. Astrocytic CO production then dilates cerebral arterioles.

This group also utilized Western blotting and immunohistochemistry to demonstrate that newborn porcine cortical astrocytes express the hydrogen sulfide-producing enzyme cystathionine β-synthase, which is important for cysteine synthesis [39].

### 2.5. Blood–Brain Barrier Modeling

Several different groups have published protocols outlining the isolation of cerebral astrocytes from the adult porcine brain [40,41,42]. These cells exhibit characteristic astrocyte morphology under phase microscopy, with multiple star-like processes extending from the cell body and uniformly expressed GFAP [41]. Cultured porcine astrocytes also express aquaporin 4 (AQP4), enhance neuronal survival, and demonstrate a dose-dependent loss of GFAP when exposed to sera from patients with Neuromyelitis optica, which is a disease characterized by astrocyte loss [42].

The Bobilya group and others have demonstrated that astrocytes can be cultured together with porcine brain capillary endothelial cells to create an in vitro blood–brain barrier model [40,43,44]. In a contact culture system, astrocytes form a confluent layer underneath the endothelial cells, and their presence in culture increases the transendothelial electrical resistance (TEER) up to nine times compared with culturing of endothelial cells alone [40]. In a non-contact culture system, astrocyte presence also contributes to increased TEER and barrier tightness [43,44] and influences endothelial cell expression of tight junction proteins [44]. The Moos lab has also demonstrated that porcine astrocytes can be cultured together with porcine brain endothelial cells and porcine pericytes in a triple culture model of the blood–brain barrier [45]. Here, they show that astrocyte presence in the tri-culture model contributes to high TEER and low permeability of endothelial cells.

### 2.6. Olfactory Bulb

The Osterberg lab published a detailed report about the structure of the adult porcine olfactory bulb (OB) [46]. Using GFAP immunohistochemistry, they reported that the porcine OB is densely populated by astrocytes, particularly in deep layers, including the granule cell layer and the mitral cell layer. The anterior olfactory nucleus (including the pars externa and pars principalis) was also populated with GFAP+ astrocytes, with the lowest astrocyte density seen in layer 2 of the pars principalis. The lateral olfactory tract was also found to be densely populated with GFAP+ astrocytes.

### 2.7. Optic Nerve and Retina

The Eppenberger group utilized GFAP immunohistochemistry and transmission electron microscopy to examine the astrocyte morphology in the porcine retina [47]. They demonstrated that GFAP staining is only located in the nerve fiber layer of the retina. Astrocytes extend an elaborate network of processes across the entire retina. Astrocytes are closely associated with blood vessels, particularly superficial blood vessels. Astrocytes were often seen wrapping their processes entirely around blood vessels, always with a higher quantity of GFAP+ processes located on the lateral/vitreal side and fewer on the scleral side of the vessels. The asymmetry with which astrocytes wrapped their processes around vessels was clearly visible with electron microscopy imaging. The authors describe astrocyte presence in the retina as forming a scaffold around blood vessels that bulge into the vitreous body.

Noda and colleagues described the expression of myocilin, the product of the gene MYOC/TIGR that is responsible for the pathogenesis of primary open-angle glaucoma in astrocytes of the optic nerve head [48]. Utilizing electron microscopy and GFAP and myocilin immunohistochemistry, they reported that myocilin was expressed in optic nerve astrocytes in the perinuclear region (outer nuclear membrane), pericentriolar region, glial filament, mitochondrial membrane, rough endoplasmic reticulum, and process endfeet near blood vessel walls. Myocilin was often found to be associated with microtubules and was also observed in astrocytes located in the lamina cribrosa region. The authors hypothesize that myocilin is an astrocyte membrane-associated cytoskeletal-linking protein that supports astrocyte cell shape and that dysregulation of myocilin in glaucoma may lead to structural alterations of the optic nerve head. This group published another report which showed that the amino acid sequence of myocilin expressed in porcine optic nerve head astrocytes is 82% homologous to human myocilin [49]. They also showed that optineurin, another glaucoma-associated gene, is expressed in porcine optic nerve head astrocytes and has an amino acid sequence that is 84% homologous with the human optineurin sequence.

Ripodas and colleagues reported that GFAP+ astrocytes in the inner layers of the porcine retina (nerve cell layer and ganglion cell layer) extend processes that make contact with blood vessels [50]. These astrocyte cell bodies and vasculature-contacting processes stain positive for vasoconstrictor endothelin-1 (ET-1), indicating that astrocytes may play a role in vascular regulation within the porcine retina.

Lee and colleagues reported the presence of heat shock protein 27 (HSP27) in the porcine retina [51]. In the adult (6 months) pig retina, they found expression of HSP27 on GFAP+ astrocytes in the ganglion cell layer and inner nuclear layer. In the newborn (postnatal day 1) pig retina, HSP27 expression was found on GFAP+ astrocytes in the ganglion cell layer.

Carreras and colleagues used immunohistochemistry and transmission electron microscopy to study the astrocyte cell–cell adhesions in the prelaminar region of the optic nerve head (ONH) [52]. They reported that the incomplete inner limiting membrane of Elschnig, which separates the ONH from the vitreous fluid of the eye, is covered with an interwoven expansion of GFAP+ astrocytes. Astrocytes closer to the vitreous stained more intensely for GFAP compared to astrocytes further away from the vitreous. However, the anterior surface of the ONH was only partially covered with astrocytes. Astrocytes at the surface of the vitreous exhibited cellular expansions that were conjoined by intercellular adherens junctions (zonulae adhesions) and occasionally gap junctions. These cells expressed calcium-dependent adhesion molecules neural cadherin (N-cadherin) in zonula adherens junctions and expressed neural cell adhesion molecules (N-CAM) in areas of cellular adhesions that were not junctional. These results indicate that astrocyte intercellular adhesions in the prelaminar region of the ONH are calcium-dependent.

Carreras and colleagues used GFAP immunohistochemistry, transmission electron microscopy, and perfusion with a fluorescent tracer to examine the pathways of fluid exchange between astrocytes within the prelaminar tissue of the porcine ONH [53]. They described the anterior interface of the ONH contacting the vitreous fluid is covered by a thick, uneven layer of astrocytes. Astrocytes expand in an interwoven fashion across this region and wrap around axons and blood vessels. The retina and optic nerve disc show extracellular spaces between the astrocyte networks that can be permeated by fluid. Astrocytes surrounding the meniscus of Kuhnt in the optic disc had large intracellular spaces and were organized together in a way the authors described as a “foamy” appearance. Astrocytes in the vitreous surface of the optic nerve are organized into an ordered “cobblestone-like” pattern. The distribution of dye across the optic disc was not uniform due to the structure of astrocyte processes, which had non-uniform, variably sized interconnected spaces between them. These extracellular spaces between astrocyte processes connected to form cavities which appeared to be preferred fluid flow routes to rid the prelaminar tissue of extra fluid.

Carreras and colleagues also published a report detailing the expression of glucose transporters in the ONH [54]. They described the ONH as having strong GFAP staining and columns of astrocytes alternating with columns of axonal bundles. GLUT1 was expressed in astrocyte endfeet, somas, and processes wrapping around axons in the ONH. They reported GLUT1 expression in the astrocyte endfeet of the nerve fiber layer, in the membranes of perivascular astrocytes of the optic disc, and in the astrocyte columns of the prelaminar regions.

Balaratnasingam and colleagues detailed astrocyte distribution in the lamina cribrosa, pre-laminar region, and post-laminar region of the porcine ONH [55,56,57,58]. Their results showed that astrocytes are the predominant glial cell in the optic nerve and that astrocytes in the ONH had radial, inter-digitating processes [56,57]. GFAP+ astrocytes in the nerve fiber layer followed the course and direction of axons, with their processes forming bundles that resembled retinal ganglion cell axons [55]. They found an absence of GFAP staining in tissue that contained laminar plates [57]. The area occupied by GFAP+ astrocytes was significantly greater in the pre-laminar region compared to the post-laminar region and the lamina cribrosa, indicating that the pre-laminar region may have the greatest metabolic demand out of these three areas [57,58]. In the pre-laminar region, GFAP staining was seen in areas where neuronal staining was present, in spaces around axonal bundles, and along the inner limiting membrane [56]. GFAP astrocytes that were present in the lamina cribrosa and post-laminar regions were also seen to closely associate with axonal bundles [56]. Additionally, higher regions of astrocyte density correlated to higher regions of axonal density in the ONH [58].

Kimball and colleagues reported on the co-expression of GFAP and the astrocyte aquaporin (AQP) channels AQP1, AQP4, and AQP9 in the ONH, retina, and myelinated optic nerve (MON) of the porcine eye [59]. They found AQP4 expression in the retinal nerve fiber layer, prelamina, and MON regions. Of particular note, they found that, similar to humans, there was no AQP4 expression in the lamina cribrosa region. There was AQP1 and AQP9 expression in the internal limiting membrane. There was no AQP1 or AQP9 expression in the lamina cribrosa or the MON. There was GFAP expression in the prelamina, lamina, lamina cribrosa, and MON. These results demonstrate that AQP4, which is normally expressed by healthy astrocytes, is not expressed in astrocytes of the lamina cribrosa of the porcine eye providing additional evidence supporting astrocytic porcine homology to humans.

Ederra and colleagues reported on the expression of the high-affinity nerve growth factor (NGF) receptor tyrosine kinase A (TrkA) in the porcine retina [60]. They performed GFAP and TrkA immunohistochemistry in the retinas of six adult porcine eyes and reported that the distribution of astrocytes in the porcine retina mimics that of the human retina. The highest quantity of GFAP+ astrocytes was found in the nerve fiber layer and ganglion cell layer of the retina, with fewer GFAP+ astrocytes found in the inner plexiform layer and inner nuclear layer. All astrocytes appeared to be closely associated with blood vessels. Most GFAP+ astrocytes in the porcine retina expressed TrkA receptors. There was a subpopulation of GFAP+ astrocytes, some of which were seen in the inner nuclear layer, that did not express TrkA receptors. The results presented in this study demonstrate that most astrocytes in the porcine retina likely play a role in modulating NGF levels.

## 3. Previous Investigations of Porcine Astrocytes in Neurotrauma Research

Various research groups have conducted examinations of astrocyte response to injury in a range of porcine models. While undoubtedly not exhaustive, our literature review is extensive, and we summarize the number of studies in each porcine injury model that investigated astrocytes to provide a glimpse at where the focus has been thus far (Figure 2). These studies and their effects on astrocytes in the porcine brain are reviewed below and summarized in Appendix A. Most of the research articles discussed below contain details about astrocyte response to injury as well as injury response that is not astrocyte related. Here, we only review the details of these papers that are related to astrocytes specifically. We omit review of other non-astrocyte, injury-associated data that are described in these articles.

### 3.1. Hypoxia–Ischemia

Several research groups have investigated astrocyte response to injury in the porcine brain using models of hypoxia (oxygen availability deficiency) and/or ischemia (blood supply restriction). The Traystman lab utilized a piglet model of asphyxic cardiac arrest to examine striatal astrocyte response at 24-, 48-, and 96 h following hypoxic–ischemic (H-I) injury [61]. They reported astrocyte degeneration at early time points (24 h and 48 h) post-H-I, seen as astrocyte cytoplasmic swelling, distended processes, loss of GFAP staining, cell death, and fragmented DNA evidenced by co-labeling of TUNEL and GFAP. GFAP cell proliferation returned to normal levels at 96 h. They additionally reported changes in astrocytic GLT-1 expression following H-I injury. While astrocyte processes primarily stained for GLT-1 in control brains, astrocyte cell bodies (instead of processes) stained positive for GLT-1 at 24 h and 48 h, and astrocytes were virtually absent of GLT-1 staining at 96h following H-I. The authors hypothesize that rapid glutamate toxicity in the striatum following H-I could be responsible for these astrocytic changes. This group then followed up by investigating whether N-methyl-D-aspartate (NMDA) glutamatergic receptors are responsible for this striatal excitotoxicity following H-I [62]. They reported that astrocytes express elevated levels of the NMDA receptor subunit NR2B in the putamen at 24 h (but not at 3, 6, or 12 h) following H-I, but that elevated expression of this receptor subunit specifically did not correlate with striatal neuronal injury resulting from H-I.

Lee and colleagues examined whether H-I injury, with and without therapeutic hypothermia and rewarming, affected the expression of endoplasmic reticulum to nucleus signaling-1 protein (ERN1) expression in astrocytes [63]. ERN1 is a marker of unfolded protein response (UPR) activation which may contribute to cell death that occurs during the cellular stress caused by H-I injury, hypothermia, and rewarming (all independent cellular stressors). While endoplasmic reticulum stress-induced activation of UPR can be neuroprotective, it can also become maladaptive. They report that hypothermia and rewarming increase the quantity of ERN1+ astrocytes in the cerebral cortex and subcortical white matter of the motor gyrus and that this change occurs in the absence of H-I injury. They further report that H-I injury followed by hypothermia reduces the quantity of ERN1+ astrocytes. There was a correlation between ERN1+ astrocytes and astrocyte apoptosis in white matter after hypothermia and rewarming, but this correlation was not seen in animals also exposed to H-I. The authors conclude that hypothermia and rewarming cause astrocyte ER stress, as evidenced by UPR activation, leading to astrocyte apoptosis. They further conclude that H-I injury might interrupt this astrocyte ER stress response that is induced by hypothermia.

Lee and colleagues also examined changes in astrocyte morphology and the correlation of these changes to fractional anisotropy (FA) from diffusion tensor imaging (DTI) following H-I injury and excitotoxic injury caused by striatal quinolinic acid (QA) injection [64]. They examined astrocyte swelling as well as expression of astrocyte cytoskeletal (GFAP), glutamate reuptake (GLT-1), and water regulation (aquaporin 4/AQP4) markers, as well as the relationship of these markers to FA measurements during DTI. They reported that QA-induced excitotoxic injury causes swollen astrocytes that display AQP4+ aggregates, cytoplasmic swelling, and vacuoles, as well as degenerating astrocytes with fragmented processes in the putamen. H-I injury similarly causes swollen astrocytes in the caudate and degenerating astrocytes with fragmented processes in the putamen. QA and H-I injury both induce swollen GLT-1+ astrocytes in the putamen. Across both types of injury, swollen GFAP+ AQP4+ GLT-1+ astrocytes seen in the caudate and putamen correlate with lower FA measured during DTI. These results indicate that swelling of astrocytes that regulated water (AQP4+) and glutamate reuptake (GLT-1+) are associated with FA changes. The authors are unsure if astrocyte swelling directly or indirectly affects FA.

The Pow lab described the altered expression of GFAP and GLAST following hypoxia in the 1-day-old piglet brain [27]. They reported that GFAP expression was upregulated in gray matter areas, including the dentate gyrus of the hippocampus, some regions of the thalamus, and some outer cortical layers. GLAST expression following hypoxia remained high in these same regions and co-localized with GFAP. This colocalization simply served to confirm that the GLAST was, in fact, astrocytic and does not indicate protein–protein interactions. Astrocytes in these regions displayed typical morphology with co-localization of GFAP and GLAST in their distal processes. Following hypoxia, GFAP expression was minimal in area CA1 of the hippocampus. There was a significant loss of GLAST expression in area CA1 of the hippocampus and cortical layers 2–5. Subregions of the thalamus also lost GLAST expression following hypoxia. Astrocyte morphology was altered in regions damaged by hypoxia, with retracted processes and GFAP and GLAST expression restricted to the soma and proximal processes rather than expressed in the distal processes. These results support this group’s hypothesis that GFAP and GLAST are co-localized such that GFAP helps to anchor GLAST in the astrocyte plasma membrane.

The Pow lab also examined the expression of the exon-9 skipping form of the glutamate-aspartate transporter EAAT1 (GLAST1b) following hypoxic injury to piglets [65]. They reported that expression of glutamate transporter 1 (GLT1a) decreases while expression of GLAST1b increases in the hippocampus (particularly the CA1 region) following H-I injury. They further report that some GFAP+ astrocytes co-localize with this GLAST1b following H-I but that most of the GLAST1b+ cells are MAP2+ neurons. The authors indicate that GLAST1b could be utilized as an important marker of neuronal damage following excitotoxic injuries. This group also reported no difference in GLAST1c expression as assessed by mRNA and protein expression in the piglet brain following hypoxia [28].

This same group then published a detailed article describing astrocyte structural changes following H-I injury in the newborn piglet brain [66]. By injecting Lucifer Yellow (LY) into fixed tissue and performing Golgi–Kopsch staining, they found that astrocytes from control subjects had fine, highly structured processes extending from astrocyte cell bodies and that multiple processes often extended from the cell to create complex arbors of processes. This contrasted astrocytes in damaged cortical gray matter regions following H-I, which had thicker processes, shorter processes, fewer processes, less complex branching structures, and often contained bulb-like swellings both along the processes and at the process terminals. Astrocytes in H-I injured animals had drastically fewer secondary and tertiary processes (i.e., processes extending from primary processes that extended directly from the soma) compared to the control. They performed Sholl analysis which revealed that astrocytes had far less branching complexity at 5–20 microns from the cell body compared to the control, a difference that was not found further out at 25–40 microns from the cell body. They also reported that astrocyte soma size significantly increased following H-I injury. These changes in astrocyte morphology occurred as early as 8 h following H-I injury and continued to become more abnormal at 72 h post-injury. This contrasted with neuronal injury, which was not observed at the 8 h time point. They reported that, in control subjects, astrocytes were of different sizes in different cortical layers and that H-I caused a decrease in astrocyte size across all cortical layers. They also performed D-aspartate uptake studies prior to fixation to examine changes in glutamate uptake in injured astrocytes. They reported that astrocytes in control brains uptake D-aspartate abundantly in their processes, whereas uptake was drastically reduced and restricted to uptake in astrocyte cell bodies at 8 and 72 h following H-I. These results collectively indicate that astrocytes in the cortex structurally and functionally change following H-I and that these astrocytic changes occur early and before neuronal changes. The authors speculate that astrocytes retract their processes quickly following injury, which damages neurons which in turn causes astrocyte proliferation and gliosis to re-establish connectivity within void brain tissue. The retraction of astrocyte processes may also lead to the observed phenomenon of glutamate transporters being closer to astrocyte cell bodies following H-I compared to their normal location along astrocyte processes. The reduced ability of astrocytes to uptake glutamate following H-I suggests that glutamate transporters may be reduced and/or distributed, and this indicates that glutamate-mediated excitotoxicity may play an important role in damage resulting from H-I injury. They also reported on changes in white matter astrocytes following H-I injury in newborn pig brains [67]. Utilizing GFAP immunolabeling, they reported that approximately 40% of cell bodies in subcortical white matter are GFAP+ astrocytes. They found reduced GFAP expression in the subcortical white matter following H-I, with the average area of GFAP+ astrocytes decreasing by 46% compared to the control. Utilizing Golgi–Kopsch staining, they also reported that white matter astrocytes were, on average, 34% smaller following H-I and that these cells had fewer processes that were shorter, thicker, and had abnormal swellings compared to control subcortical white matter astrocytes.

This group then published another report detailing changes in the presence of glutamine synthetase (GS), the glutamate detoxification enzyme that converts ammonia and glutamate into glutamine, in a newborn porcine model of H-I injury [68]. They reported changes in GS as soon as 1 h following H-I injury, with small patches of GS-devoid regions seen in regions of the cortex and CA1. At 24 and 72 h following H-I, they report extensive areas of GS loss in the cortex and CA1 (areas generally vulnerable to H-I injury) but not in the dentate gyrus or thalamus (areas not typically vulnerable to H-I injury). Astrocyte GS loss at these later time points overlapped with astrocyte GLAST loss. Like this group’s previous work, these results suggest that astrocytes respond early to H-I injury, evidenced here by the early loss of GS that continues to become more pronounced over time. They argue that this early loss of GS likely leads to an accumulation of glutamate in astrocytes which would reduce their ability to uptake glutamate and, in turn, exacerbate the potential for glutamate toxicity following injury. Reduced GS may also lead to ammonia accumulation in the brain which would greatly disrupt ammonia homeostasis.

This group also investigated the presence of two phosphorylated GFAP proteins, p8GFAP and p13GFAP, following H-I injury in newborn piglets [69]. They reported that, even in control brains, pGFAP is expressed in astrocytes with normal morphology. Astrocytes that expressed pGFAP lacked the fine and bushy processes of normal astrocytes and instead had short processes and thickened varicose processes. Specifically, astrocytes expressed pGFAP in processes near the cell body, in abnormal terminal dilations on their processes, and in endfeet contacting blood vessels. They also reported upregulated pGFAP expression in brain regions that were injured following H-I. p8GFAP was upregulated in the cortex at 24 h following H-I and was upregulated in the cortex, basal ganglia, and thalamus at 72 h following H-I. p13GFAP expression did not change at 24 h following H-I but was upregulated in the cortex and basal ganglia at 72 h following H-I. They also reported that higher pGFAP expression correlated with higher histological injury. Overall, pGFAP expression appears in astrocytes with abnormal morphology and is upregulated in injured regions following H-I. The authors speculate that phosphorylation of GFAP, despite being an energy-dependent mechanism, could be part of a repair mechanism following injury.

Ruzafa and colleagues exposed newborn pigs to hypoxia and then examined astrocyte changes in the retina and superior colliculus at 4 h following injury [70]. They reported that astrocyte networks in the retina, which typically run parallel to retinal ganglion cell axons, appeared more disorganized following hypoxia. Although these differences in organization were not significant, they reported more laterally extending astrocyte processes and a higher degree of randomness of retinal astrocytes following hypoxia compared to control. They also reported that astrocytes in the superior colliculus exhibited hypertrophy, increased density, and increased astrocyte cytoskeletal area following hypoxia. These results indicate that astrocytes in the brain may be more susceptible to early damage from hypoxia compared to astrocytes in the retina. The authors speculate that this resilience of the retina to oxygen deprivation may be due to the presence of Müller glial cells, which exist in the retina but not in the brain.

Zheng and Wang investigated changes in lactate and glucose metabolism in the basal ganglia between 2 and 72 h following hypoxic–ischemic injury in piglets [71]. In this study, they utilized H&E staining to demonstrate that astrocytes in the basal ganglia become swollen at 6 h following H-I and that these swollen astrocytes further exhibit condensed nucleoli at 24 h and become degraded at 48 h following H-I. They also reported that expression of monocarboxylate transporter 4 (MCT-4), a lactate transporter that is primarily expressed by astrocytes, is increased in the basal ganglia at 12–24 h following H-I. These results indicate that astrocytes exhibit early morphological changes following H-I and that the lactate metabolism of astrocytes is altered following H-I.

Parfenova and colleagues investigated the role of astrocyte-produced carbon monoxide in both in vitro porcine astrocyte cultures and an in vivo model of neonatal asphyxia in the piglet brain [72]. They report that astrocyte cultures highly express heme oxygenase 2 (HO-2) and produce CO in the presence of the prooxidants glutamate and TNF-alpha. They also demonstrate that in vitro and in vitro porcine astrocytes drastically increase CO production (6–10-fold) when exposed to asphyxic conditions. Asphyxia also leads to increased astrocyte ROS production in vivo, which is further increased when subjects are pre-treated with the HO inhibitor SnPP and are decreased when subjects are pre-treated with the CO-releasing molecule A1 (CORM-A1) and bilirubin. Astrocyte cultures that were exposed to excitotoxic glutamate had increased ROS production and increased apoptosis, and this was augmented when astrocyte CO production was inhibited by SnPP. They also demonstrated that pial responses to the astrocyte-dependent vasodilators ADP and glutamate were compromised at 24 and 48 h post-asphyxia. Pial responses to ADP and glutamate were further inhibited when subjects were pretreated with SnPP prior to asphyxia. On the other hand, the reduced pial responses to ADP and glutamate resulting from asphyxia were prevented by pretreatment with CORM-A1. Overall, these results demonstrate that cortical astrocytes in the porcine brain respond to asphyxia by activating HO-2 and increasing the production of CO, which acts as an antioxidant and cytoprotective messenger against oxidative stress induced by asphyxia. These cytoprotective effects are increased with CO levels and augmented and reduced when HO is inhibited. These results indicate that CO donors could be an effective approach to treating prolonged asphyxia.

### 3.2. Middle Cerebral Artery Occlusion

Spellicy and colleagues recently published an article detailing their use of high-content image (HCI) analysis to examine changes in astrocyte morphology following middle cerebral artery occlusion (MCAO) in the adult Yucatan porcine brain [73]. Four weeks post-stroke, they performed GFAP immunohistochemistry and utilized HCI to examine 19 astrocyte morphological parameters in the perilesional area and ipsilateral hemisphere of stroke and non-stroke subjects. These methods provided a wealth of information on subtle changes in astrocyte morphology following injury. They reported larger, more extended, more ramified astrocytes in the perilesional area and ipsilateral hemisphere following stroke, compared to smaller, more rounded astrocytes in non-stroke subjects. Additionally, they reported significant increases in GFAP+ area shape perimeter, major axis length, and mean radius and a significant decrease in solidity in the ipsilateral hemisphere stroke subjects compared to non-stroke subjects. In the perilesional area, they reported significant increases in GFAP+ area, compactness, and major axis length and significant decreases in form factor, solidity, and orientation in stroke subjects compared to non-stroke subjects. Overall, their analysis indicated a higher GFAP+ area and more reactive astrocyte morphology following stroke. The semi-automated analyses employed in this article provided an in-depth analysis of changes in astrocyte morphology following MCAO.

### 3.3. Controlled Cortical Impact Brain Injury

Baker and colleagues developed a graded cortical control impact (CCI) brain injury model in 3-week-old piglets [74]. They described the effect of CCI on astrocytes by comparing GFAP reactivity in the perilesional area compared to the same area in the contralateral, uninjured hemisphere. They reported that increased GFAP reactivity in the perilesional area (compared to the contralateral side) correlates with increased CCI severity (increased impact velocity and impact depth). As CCI injury increased, the quantity and intensity of astrocytes in the perilesional area increased, indicating increased astrocyte proliferation (astrocytosis) and hypertrophy (astrogliosis) with more severe injury. The authors reported that augmented GFAP expression likely results from increased astrocyte size and proliferation following injury, as well as astrocyte migration toward the injury site.

### 3.4. Fluid Percussion Brain Injury

Lafrenaye and colleagues performed central fluid percussion injury (cFPI) in Yucatan mini pigs and collected blood samples at various time points up to 6 h post-injury [75]. They found steadily increasing blood serum levels of GFAP as time following injury increased, with significantly greater serum levels at 3 and 6 h post-injury compared to sham. Upon examining GFAP immunohistochemistry in the thalamus at 6 h post-injury, they reported that astrocyte GFAP intensity negatively correlated with cell area and cell roundness. These morphological changes are indicative of injured astrocytes. They also found more subtle differences in astrocyte morphology by examining GFAP+ cells in the thalamus using ultrastructural imaging. For example, GFAP+ astrocytic endfeet appeared more pronounced around vessels following cFPI injury compared to sham. Additionally, they discovered GFAP+ vesicles in the basement membrane and endothelial cell cytoplasm following cFPI injury. These findings indicate a possible mechanism for vesicular GFAP transport directly from astrocyte endfeet through endothelial cells lining blood vessel walls and into the blood. Additionally, these studies demonstrate that circulating serum biomarkers can be useful metrics to assess subtle changes in histopathology following diffuse brain injury.

### 3.5. Head Rotational Acceleration Injury

Currently, the only preclinical model that recreates the mechanisms and manifestations of human TBI relies on head rotational acceleration injury in pigs [14]. The core acceleration-induced forces exerted throughout the brain during human TBI are dependent on brain mass, which is why attempts to replicate acceleration-based injury in rodents with the CHIMERA model fail to reach scaled thresholds [76,77,78,79,80]. Pigs possess brains of sufficient mass to replicate human TBI forces by scaling up acceleration, and recreating this core mechanism of human TBI also recreates key manifestations of human TBI that do not appear in any other models, such as loss of consciousness [23,24,81]. Three previous studies utilizing this model reported from the Smith, Meaney, and Cullen labs include information on astrocyte response to rotational acceleration injury. In 1997, Smith and colleagues induced rotational acceleration injury in Hanford swine in the coronal plane and reported reactive astrocytosis in the molecular layer of the cerebral cortex, in subcortical white matter regions, in the corpus callosum, and in the CA1 and CA3 regions of the hippocampus [82]. Reactive astrocytosis was reported to be much greater in the hippocampus compared to other regions. Twenty years later, Johnson and colleagues investigated BBB disruption and associated astrocyte response at 6 h, 48 h, and 72 h following coronal head rotational acceleration in Hanford swine [83]. In addition to demonstrating BBB disruption via leaked serum proteins fibrinogen (FBG) and immunoglobulin (IgG), they report that these serum proteins are internalized by surrounding astrocytes in the brain parenchyma, as evidenced by GFAP-positive cells with astrocyte morphology surrounding FBG or IgG immunoreactivity (Figure 3). While the number of astrocytes internalizing these serum proteins was not quantified, striking images clearly demonstrate co-localization of GFAP with these serum proteins in the premotor cortex and parietal cortex at 72 h post-injury. The Cullen Lab then also conducted studies that included analysis of astrocyte response to head rotational acceleration injury. Grovola et al. analyzed astrocyte reactivity via GFAP expression following two severities of mild injury in the coronal plane [84]. They scored GFAP+ cell size and the density of GFAP+ cells in the hippocampus, periventricular white matter, inferior temporal gyrus, and cingulate gyrus and averaged these scores to provide a single score of reactivity for each subject. They reported no change in astrocyte size or density in either injury condition. Unfortunately, astrocytes have been mostly overlooked in this highly translational model, limited to GFAP-based morphological assessments after mild TBI, and deeper study is required.

### 3.6. Explosive Blast Injury

To our knowledge, just three previous reports on the effects of explosive blast injury in the porcine brain have included an analysis of astrocyte response to this injury. In 2011, explosive blast injury was performed on Yorkshire swine to simulate a blast experienced in an open field (blast tube), tactile vehicle, or a building [85]. Seventy-two hours and two weeks following injury, GFAP+ cells were found to be greater in the hippocampus, corpus callosum, parasagittal cortex including cingulate gyrus, and superior, middle, and inferior frontal gyri. They reported increased cell density in the hilus and molecular layers of the hippocampus. Interestingly, despite the reported increase in GFAP+ astrocytes, the cells remained within distinct domains and did not display the phenotype typically characteristic of injured astrocytes. The authors speculate that the augmented astrocyte numbers combined with the normal astrocyte phenotype could indicate that astrogliosis resulted from proinflammatory factor release from activated microglia or transient blood–brain barrier permeability rather than resulting directly from neuronal injury. A few years later, this research group investigated the effects of a single, double, or triple blast on astrocyte proliferation and reactivity in Yucatan minipigs [86]. They found augmented astrocyte density in the dentate hilus and molecular layer of the hippocampus following blast injury (single, double, and triple injury) compared to sham. This astrocyte proliferation was found between 2 weeks and 6–8 months following blast exposure. Furthermore, this increase in astrocyte density was greater in animals exposed to two and three blasts compared to one blast. Examined astrocytes had a stellate appearance and non-overlapping domains. Additionally, animals exposed to three blasts also had astrocyte proliferation in the deep central white matter. The researchers speculate that astrocyte activation may be an early hippocampal response to injury. A different research group investigated the effects of open field blast exposure (medium and high blast overpressure) on astrocyte proliferation at 3 days following blast exposure [87]. They observed significantly higher quantities of astrocytes in frontal lobe regions in both the medium and high blast groups compared to sham animals, with most astrocytes seen in white matter. There were also more astrocytes in the high-blast group compared to the medium-blast group. The observed astrocytes had enlarged cell bodies and processes and more intense GFAP staining compared to sham, although these observations were not quantified. Overall, the above reports reveal that explosive blast injury causes astrocyte proliferation in all reported studies ranging from 3 days to 8 months following blast exposure.

### 3.7. Cerebral Edema via Water Intoxication

The Rosenthal group performed water intoxication in female swine as a mechanism to investigate cerebral edema leading to elevated intracranial pressure [88]. They took electron microscopy samples of brain tissue (harvested from over the left coronal suture 1 cm lateral to the midline) before and after serially inducing four levels of water intoxication in each subject. While no widespread brain astrocyte pathology was examined, transmission electron micrographs revealed substantial swelling of astrocyte endfeet in injured tissue when compared to tissue samples taken prior to injury. These swollen astrocyte endfeet were found around capillary endothelial cells.

### 3.8. Intracerebral Hemorrhage via Blood Injection

Zhou and colleagues induced intracerebral hemorrhage in adult male pigs by injecting blood into the frontal lobe to study the expression of the integrin-associated protein cluster of differentiation 47 (CD47) following injury [89]. CD47 plays a critical role in immune responses by providing inhibitory signals to phagocytic processes regulating inflammatory and immune responses. They reported that while CD47 was increased throughout the brain, this protein did not co-localize with astrocytes (defined as GFAP+ cells) in the perihematomal region 3 days following injury. These results indicate that astrocytes may not be directly involved in the increase in brain CD47 expression following injury.

### 3.9. Multiple Trauma and Hemorrhagic Shock

Vogt and colleagues studied the effects of multiple trauma associated with hemorrhagic shock (MT/HS) in male pigs [90]. They performed two different severities of MT/HS injury: MT with 45% blood loss and 90 min HS phase (T90) and MT with 50% blood loss and 120 min HS phase (T120). In separate groups for both injury severities, they induced hypothermia (TH90, TH120) to examine how this treatment might change the analyzed injury effects. They analyzed the levels of the calcium-binding astrocytic protein S100B at six time points up to 48.5 h following trauma induction and analyzed S100B immunohistochemistry at 48.5 h following trauma induction. They reported that S100B blood serum values were transiently increased in the T120 group compared to sham and that hypothermia did not influence serum S100B levels compared to normothermic groups across both injury conditions. Although they did not report any change between groups in S100B+ astrocyte cell count in the frontal lobe, they did note that S100B+ astrocytes in all four trauma groups had enlarged somata and elongated branches compared to astrocytes in control and sham groups. This lack of astrocyte proliferation following MT/HS injury caused authors to speculate that the temporary increase in S100B serum levels likely resulted from the trauma and/or shock but not directly from cerebral damage.

### 3.10. L-2-Alpha-Aminoadipic Acid (L-AAA) Toxicity

Leffler and colleagues performed a set of experiments examining the effects of glial toxin L-AAA on pig astrocytes in culture and in vivo [91]. They reported that treating cultured porcine astrocytes with 0.2–2 mM L-AAA for 2 h dose-dependently increased cell detachment, process retraction, loss of cell–cell contacts, and cytoskeletal changes. They further reported that in vivo administration of 2 mM L-AAA for 5 h significantly disrupted the confluent layer of GFAP+ superficial glia limitans, as evidenced by GFAP immunohistochemistry. In vivo L-AAA administration additionally eliminated pial arteriolar dilation to the astrocyte-dependent dilators ADP and glutamate, as well as eliminated glutamate-stimulated CO production at the cortical surface. These results suggest that astrocytes employ CO as a mechanism to induce glutamatergic vasodilation in the cerebral cortex. The authors hypothesize that glutamate activates astrocyte glutamate receptors to stimulate CO production, which then dilates pial arterioles. A few years later, these authors published another report further elucidating this mechanism [92]. They utilize a methodology to demonstrate that injuring astrocytes via the application of 2 mM L-AAA or the heme oxygenase (HO) inhibitor CrMP to the parietal cortex in vivo blocks both pial arteriolar dilation and ADP-induced CO production. These results further confirm their previous findings that glia limitans astrocytes utilize CO as a gaseous neurotransmitter to mediate ADP-induced pial arteriolar dilation. Further, these findings suggest that CO is produced by astrocytes via a HO-catalyzed reaction. This group continues their research to investigate whether cortical astrocyte ionotropic glutamate receptors (iGluRs), specifically NMDA- and AMPA/kainate-type receptors, mediate the glutamate-induced, astrocyte-dependent HO activation that in turn causes cerebral vasodilation [93]. They demonstrate that NMDAR agonists (NMDA and cic-ACPD) and AMPA receptor agonists (AMPA and kainate) cause pial arteriolar dilation in vivo and that the glial toxin L-AAA blocks this response. They further demonstrate that in vivo application of the NMDAR antagonist D-AP5 and the AMPA/kainite receptor antagonist DNQX block pial arteriolar dilation that results from AMPA, NMDA, or glutamate application. These results demonstrate that astrocytic iGluRs play an essential role in the dilation of pial arterioles in the piglet cortex.

### 3.11. Seizure

The Parfenova lab has also published two research articles which examine the effects of cerebral astrocyte response to neonatal seizure caused by bicuculline administration [94,95]. Both studies examine the effect of seizures on cerebral vascular response to the astrocyte-dependent vasodilator ADP and endothelium and astrocyte-dependent vasodilators glutamate, the AMPA receptor agonist L-quisqualic acid, and the HO substrate heme. In addition to examining general changes in the response of pial arterioles to these vasodilators after a seizure, they also investigate whether administration of CORM-1A (a carbon monoxide-releasing molecule) can improve vascular outcome following seizure. They report that cerebral vascular response to ADP, glutamate, quisqualic acid, and heme is significantly reduced following seizure compared to control groups [94]. Administration of CORM-A1 either 10 min prior to seizure induction or 20 min after seizure induction was able to prevent this lower cerebral vascular response to ADP, glutamate, and heme [94]. These studies suggest that the astrocyte components of the neurovascular unit are injured during a seizure, which could contribute to the dysregulation of blood flow in the brain. Moreover, these studies indicate that functional astrocytes are necessary for regular cerebral blood flow. Additionally, the positive effects of CORM-1 administration on this dysregulation could indicate that carbon monoxide administration in the brain could help alleviate some of this astrocyte damage before or after seizure induction. They further report no sex differences in these findings [95].

### 3.12. Bovine Spongiform Encephalopathy

Liberski and colleagues experimentally infected porcine with bovine spongiform encephalopathy (BSE) and performed transmission electron microscopy imaging to examine the ultrastructural pathology caused by this neurodegenerative disorder [96]. They reported that BSE causes astrocytosis and astrocyte processes to be in close conjunction with microglial cells. These ultrastructural findings parallel previous reports of cattle infected with BSE and humans infected with transmissible spongiform encephalopathies (TSEs).

### 3.13. Mechanical Ventilation

The Reynolds research group investigated brain injury caused by lung-protective mechanical ventilation in a porcine model [97]. Following 50 h of mechanical ventilation (MV), subjects had a higher percentage of GFAP-positive reactive astrocytes in the hippocampus compared to never-ventilated (NV) subjects. Astrocyte proliferation in the hippocampus indicates the hippocampus is damaged because of MV. Additionally, MV subjects had higher blood serum levels of GFAP compared to NV subjects, indicating astrocyte damage. The authors speculated that elevated serum GFAP levels could be caused either directly by lung injury or by a brain injury that occurred during MV. These researchers additionally reported that transvenous diaphragm neurostimulation (TTDN) during MV can reduce these negative effects [98]. Furthermore, neuroprotection was greater in subjects exposed to TTDN during every breath during MV compared to subjects exposed to TTDN every other breath during MV. The authors speculate that by mimicking the pulmonary stretch receptor response that occurs during spontaneous breathing, TTDN during MV may regulate dopamine release in the hippocampus to offer neuroprotection.

### 3.14. Cardiac Arrest and Cardiopulmonary Bypass (CPB)

Sharma et al. examined changes in GFAP expression and astrocyte morphology via transmission electron microscopy following 12 min of untreated cardiac arrest (ventricular fibrillation) in piglets [99]. They reported augmented GFAP reactivity in both the thalamus and cerebral cortex at 30 and 60 min following injury, with GFAP reactivity correlating well with neuronal damage and albumin leakage. Immunohistochemistry revealed an increase in star-shaped astrocytes around blood vessels and neurons, and electron microscopy revealed swollen astrocytes in the thalamus and cortex at 60 min following cardiac arrest. Overall, this study demonstrated astrocyte reactivity and morphological changes in diverse brain regions at early time points following cardiac arrest.

In 1989, Laursen and colleagues examined GFAP expression following total cardiopulmonary bypass (CPB) for two hours in pigs [100]. They did not observe any astrocytosis following CPB. However, they did observe perivascular swelling of astrocyte endfeet in white and gray matter following CPB that occurred at normothermia. This swelling of astrocytic endfeet was prevented when CPB was conducted at hypothermia. The authors speculate that astrocyte endfeet swelling could have been caused by a relative impairment of Na+-K+ exchange.

Around 30 years later, Stinnett and colleagues examined changes in fractional anisotropy (FA) obtained from diffusion tensor imaging and the correlation of these changes to changes in GS+GFAP+ astrocyte numbers following mild and severe CPB in piglets [101]. They reported acute astrogliosis following severe CPB and no differences in astrocyte quantities between sham, mild, and severe CPB injury by 4 weeks post-injury. They also reported that FA was not associated with astrocyte quantity in control subjects but that FA changes were positively correlated with astrocyte numbers in the acute postoperative period following CPB. Collectively, their results show that the quantity of white matter astrocytes changes acutely following CPB and that these changes can be captured with FA.

### 3.15. Sepsis

Papadopolous and colleagues utilized electron microscopy to explore astrocyte morphology changes in the cortex following sepsis induced by fecal peritonitis in adolescent pigs [102]. They reported that astrocyte endfeet were occasionally swollen in sham subjects. In contrast, astrocyte endfeet were frequently and intensely swollen in septic subjects. Astrocyte endfeet in septic subjects also presented ruptured perimicrovessel membranes. The authors speculate that injury to astrocytic endfeet caused by sepsis could potentially impair astrocytic metabolic activity.

### 3.16. Hepatic Clamping and Hepatic Encephalopathy

Diemer and Tonnesen investigated glial changes following portocaval anastomosis (PCA) and either total or partial hepatic artery clamping/devascularization [103]. Utilizing H&E staining, they reported no astrocyte changes following PCA and total hepatic devascularization. They found increased density of astrocyte nuclei and increased astrocyte nuclei diameter in the frontal cortex and putamen following PCA and partial hepatic clamping for 30–60 min. Additionally, they found enlarged astrocytes with watery nuclei and peripheral nucleoli in the frontal cortex in the temporary hepatic clamping group, which is indicative of Alzheimer’s type II astrocytes (AIIA) (Figure 4).

Kristiansen and colleagues induced acute liver failure in pigs by placing an end-to-end portocaval shunt and ligating the hepatic arteries and analyzed the frontal lobe, pons, and cerebellum with electron microscopy imaging 8 h later [104]. They reported several ultrastructural astrocyte changes in these brain regions following ALF, including cytoplasmic swelling, increased electron density, condensation of the cytoplasm, clumping of nuclear chromatin, and cytoplasmic membrane dissolution. Electron microscopy imaging allowed this group to find several different areas of astrocyte injury following ALF that are indicative of necrotic cell death from this injury phenotype. Zeltser et al. created a model of diet-induced nonalcoholic fatty liver disease (NAFLD) in 13-day-old juvenile pigs by feeding them a high-fructose, high-fat (HFF) diet for 70 days [105]. They analyzed the GFAP intensity per area in the frontal cortex and reported increased astrogliosis in HFF subjects compared to subjects fed a control diet. They also demonstrated that GFAP intensity increased with the severity of liver disease, indicating that metabolic changes caused by NAFLD can lead to neurodegenerative changes, including astrogliosis.

Kanai and colleagues induced acute hepatic failure in male mini-pigs by administration of 0.05 mg/kg alpha-amanitin and 1 ug/kg lipopolysaccharide (LPS) in the splenic vein and investigated the effect of this injury on blood serum S100B levels with ELISA several hours after injury [106]. They reported that acute hepatic failure led to increased S100B serum protein levels and that S100B levels were particularly elevated in animals that died from induction of acute hepatic failure. They also reported that treatment of subjects with bioartificial liver (BAL) therapy for 4–6 h after hepatic failure induction decreased these plasma levels of S100B. This study reveals that acute hepatic failure induces astrocyte damage, as seen by elevated S100B serum levels, and that this astrocyte damage can be mitigated with BAL therapy.

The Cholich group induced hepatic encephalopathy by poisoning pigs with 5–10% *Senna occidentalis* (*S. occidentalis*) seeds [107]. Brain histology was studied within 12 h of the first observation of clinical symptoms, which occurred 7–11 days following poisoning. Utilizing H&E immunohistochemistry, they reported that hepatic encephalopathy subjects had AIIA in the cerebral cortex that exhibited nuclear pallor, chromatin margination, and swelling. GFAP immunohistochemistry revealed that white matter astrocytes in control subjects had long, branching processes that expressed GFAP, whereas white matter astrocytes in subjects treated with 10% S. occidentalis had reduced immunoreactivity to GFAP, cell shrinkage, and process retraction. Treated subjects had AIIA in gray matter regions that were GFAP and had overall reduced numbers of GFAP astrocytes and a reduced percentage of GFAP+ area in white matter compared to control. The authors speculated that these morphological and intermediate filament changes following poisoning with *S. occidentalis* may precede glial cell death resulting from poisoning.

### 3.17. Poisoning

Finnie and colleagues published a case report on the presence of AIIA in the brains of pigs that were subject to salt poisoning resulting from at least 2 days of water deprivation [108]. Utilizing H&E and GFAP histochemistry, they reported the presence of numerous AIIA randomly distributed in the cerebral cortical gray matter. These cells were minimal in subcortical white matter and were absent in the brains of control subjects. The morphology and appearance of these AIIA were described as swollen, clear, and watery. These cells also had scant chromatin, vacuolated nuclei, and reduced GFAP expression. They were sometimes aggregated in clusters of 2–3. The authors speculated that reduced GFAP staining in AIIA could be due to the instability of GFAP mRNA in these cells. Additionally, they hypothesized that the AIIA phenotype may be a transitionary phenotype between non-reactive astrocytes with normal GFAP expression and reactive astrocytes with augmented GFAP expression.

The Leifsson group examined changes in the astrocyte phenotype following the natural infection of pigs with the parasite *Taenia solium* (*T. solium*) [109]. They described normal astrocyte distribution in uninfected brains, with higher expression of astrocytes and astrocyte endfeet in the medulla compared to the cortex. There was an increase in astrocyte size, number, and GFAP expression in subjects infected with *T. solium*. This astrogliosis resulting from infection was more pronounced around cysticeri (larval cysts) found in the cortex compared to the medulla. They also noted that some lesions were surrounded by astrocyte endfeet forming a glial scar, while others were surrounded by blood vessels that lacked astrocyte endfeet (indicative of BBB loss). Overall, astrogliosis is prominent after *T. solium* neurocysticercosis.

Riet-Correra et al. examined the effects of ingesting 13% *Aeschynomene indica* (*A. indica*) seeds in adult pigs [110]. A. indica is a weed that grows abundantly in irrigated rice fields and can contaminate rice harvests. Utilizing H&E staining and transmission electron microscopy, they reported that astrocytes in the cerebellum were enlarged and swollen 24 h after ingestion, with endfoot processes often seen separating capillary endothelial cells and pericytes. They also reported astrocytosis in the cerebellum at 15 days following *A. indica* ingestion.

### 3.18. Intrauterine Growth Restriction

Intrauterine growth restriction (IUGR), caused by placental insufficiency, leads to fetal development in a chronic hypoxic environment that, in turn, leads to neurological disabilities. Wixey and colleagues published a report detailing the effects of IUGR on astrocyte density and morphology in postnatal-day 1 (p1) and 4 (p4) piglets [111]. They reported that GFAP+ astrocytes in the parietal white matter of normal growth (NG) brains possessed long, branching processes and small cell bodies. On the other hand, GFAP+ astrocytes in the parietal white matter of IUGR brains had morphology indicative of reactive astrocytes, including larger cell bodies and fewer, shorter, retracted processes. Analyzing GFAP+ areal density revealed that IUGR brains significantly increased GFAP+ density in intragyral white matter, subcortical white matter, and periventricular white matter compared with NG brains at both p1 and p4. Additionally, GFAP+ astrocytes co-localized with IL-1beta in periventricular white matter and with IL-18 and TNFalpha in the parietal cortex of IUGR brains. NG brains had a very minimal overlap of these inflammatory markers with GFAP+ astrocytes. Overall, these results reveal that chronic hypoxia during fetal development causes increased astrocyte density and astrocytes to possess a reactive morphology up to at least p4. This group then published two studies examining whether Ibuprofen treatment immediately after birth affected this inflammatory astrocytic morphology seen in IUGR brains [112,113]. They reported that treatment of Ibuprofen on postnatal days 1–3 alleviated the increase in GFAP+ density in intragyral white matter, subcortical white matter, and periventricular white matter that was seen in IUGR brains at p4 [112]. These results indicate that Ibuprofen treatment in the first few days after birth may hinder inflammatory mediators and thus reduce astrocyte reactivity in white matter regions. They also investigated the effects of IUGR and Ibuprofen treatment on the integrity of the neurovascular unit, including the interaction between astrocytic endfeet and blood vessels [113]. They reported that IUGR brains had significant loss of GFAP labeling around vasculature, increased hypertrophy of GFAP+ endfeet around vasculature, and decreased quantity of astrocyte endfeet interacting with the vasculature (overall loss of astrocyte coverage of vasculature). Ibuprofen treatment for three days following birth caused astrocyte interactions with vasculature to return to normal, with Ibuprofen subjects exhibiting normal astrocyte coverage of vessels and reduced endfeet hypertrophy. IUGR brains also contained GFAP+ astrocytes that took extraverted serum proteins albumin and IgG. Astrocytes co-labeling with these serum proteins tended to maintain their normal interactions with blood vessels but had very hypertrophic endfeet. Ibuprofen treatment reduced serum protein extraversion and co-labeling of astrocytes with serum proteins. IUGR brains also had increased expression of GFAP+ co-labeling with the tight junction protein claudin-1 (Cldn1) compared to NG brains, an increase that was also reduced with Ibuprofen treatment. Overall, these results indicate that the neurovascular unit is disrupted because of IUGR, including disrupted astrocyte interactions with vasculature and BBB disruption and leakage. Ibuprofen treatment decreased astrocyte reactivity and restored healthy astrocyte interactions with vasculature, helping to reduce BBB leakage. Anti-inflammatory signaling caused by Ibuprofen treatment appears to help restore BBB integrity in IUGR subjects. To our knowledge, this has not been investigated clinically.

### 3.19. Maternal Infection and Poisoning

Prenatal maternal infections can lead to fetal brain abnormalities. Antonson and colleagues investigated astrocyte changes in the Large White/Landrace porcine fetal hippocampus following maternal infection with porcine reproductive and respiratory syndrome virus (PRRSV) [114]. Pregnant female pigs were infected with PRRSV at gestational day (GD) 76, and fetuses were removed for analysis on GD111, 3 days before their expected delivery date. Using quantitative real-time PCR, they reported increased GFAP expression in the fetal hippocampus of PRRSV-infected mothers relative to fetuses of uninfected mothers. They also utilized GFAP immunohistochemistry to show that the relative integrated density of GFAP+ cells in the hilar region of the hippocampus was greater following maternal PRRSV infection compared to uninfected mothers. These results demonstrate upregulation of GFAP+ gliosis following late-gestation maternal PRRSV infection, indicating that maternal infection during pregnancy causes astrocyte responsiveness in developing fetuses.

Recently the Pankratova Lab published a study investigating the exposure of porcine fetuses to intra-amniotic injection of 1 mg liposaccharide (LPS), which can cause intrauterine inflammation at embryonic gestation day 103 (E103) prior to cesarean section delivery at E106 [115]. They analyzed GFAP immunoreactivity in the cortex, hippocampus, and periventricular white matter at day 1 (P1) and day 5 (P5) after birth. They reported no differences between control and LPS-treated subjects in GFAP immunoreactivity across these three brain regions at either P1 or P5. They did show an increase in GFAP immunoreactivity in the periventricular white matter from P1 to P5 in all subjects, reflecting a developmental increase in astrocytes during this time. GFAP did not increase from P1 to P5 across the cortex or hippocampus. The authors speculated that an increase in GFAP in the periventricular white matter specifically could be because astrocytes play an important role in myelination during this early time by supplying energy, trophic factors, and iron to oligodendrocytes and their precursors, among other supportive functions [116]. Additionally, their results showed that white matter astrocytes express overall higher levels of GFAP than gray matter astrocytes at both P1 and P5.

Brunse and colleagues investigated the effect of preterm birth, often associated with impaired neurodevelopment and necrotizing enterocolitis, on perivascular astrocyte coverage in the hippocampus and striatum of the porcine brain at 8 h and 5 days following birth [117]. Following the delivery of fetuses at preterm (106 days of gestation) and full-term (117 days of gestation), they conducted Western blotting and reported that GFAP protein levels in the hippocampus and striatum were similar between these groups at both 8 h and 5 days following birth. However, they noted that perivascular astrocyte coverage (determined by overlapping immunohistochemistry of GFAP and laminin) was three-fold higher in full-term compared to preterm pigs in the hippocampus. This difference was still found at 5 days following birth. Perivascular astrocyte coverage was similar between groups in the striatum. These results indicate that astrocyte endfeet coverage of the BBB is reduced in the hippocampus, but not the striatum, following preterm birth. This indicates that the hippocampus might be more vulnerable than the striatum to BBB disruption following preterm birth.

### 3.20. Injury to the Optic Nerve

In an earlier section, we described a publication that reported on the amino acid sequences of the glaucoma-associated genes myocilin and optineurin [49]. In this same publication, the authors examined the effects of different stressors on the level of these two genes in astrocytes isolated from the trabecular and prelaminar regions of the porcine optic nerve head. Expression of myocilin and optineurin was assessed by RT-PCR isolated from cultured cells following exposure to various stressors. Incubating porcine optic nerve head astrocytes under conditions of hydrostatic pressure (33 mg Hg above atmospheric pressure for between 12–72 h) or mechanical stretching (10% mechanical stretch over 24 h) caused no difference in myocilin or optineurin expression as analyzed via total RNA expression. Incubating porcine optic nerve head astrocytes under hypoxic conditions (7% O_2_ and 5% CO_2_ for 72 h) caused significantly decreased myocilin expression and no change in optineurin expression. Exposing porcine optic nerve head astrocytes to dexamethasone (500 nM added to culture media for 2 weeks) caused significantly decreased optineurin expression and significantly increased myocilin expression. The differing responses in expression levels of these two genes under different stressors indicate that myocilin and optineurin induce glaucoma via different mechanisms.

Balaratnasingam and colleagues examined astrocyte damage in the porcine optic nerve head following an acute increase in intraocular pressure (IOP) for 3, 6, 9, or 12 h [56]. There was no change in GFAP intensity following increased IOP for 3 h. After elevated IOP exposure for 6, 9, or 12 h, there was decreased GFAP expression in the pre-laminar, post-laminar, and lamina cribrosa regions. The percentage of ONH tissue that stained positive for GFAP decreased following 12 h of elevated IOP, a change that was not seen after shorter exposure times. They also reported that the architecture of astrocytes changed following increased IOP. Under normal conditions, GFAP+ astrocytes had a reticulated and skein organization across all regions of the ONH. Following increased IOP, astrocytes were disorganized and had round and ovoid changes to their structure. These changes were seen across all regions of the ONH (pre-laminar, post-laminar, and lamina cribrosa) but only in a subset of astrocytes. They observed patches of unaltered astrocytes interspersed with patches of morphologically altered astrocytes across these three regions following increased IOP. These morphological changes were seen in all groups. In subjects exposed to increased IOP for 12 h, the GFAP morphology changed drastically to elicit an amorphous appearance, having lost the structure of fine processes and gained nodular enlargements. In subjects exposed to elevated IOP for 3 h, morphological changes occurred in the lamina cribrosa. In subjects exposed to increased IOP for 6, 9, and 12 h, morphological changes were seen in the pre-laminar, post-laminar, and lamina cribrosa regions. The authors hypothesized that these morphological changes are due to astrocyte swelling and likely result from cell injury rather than cell reactivity.

This group also reported astrocyte changes following argon laser-induced axotomy of the porcine retinal ganglion cell axon [55]. A reduction in the intensity of GFAP staining was observed at the axotomized region as well as up to 2400 microns on the peripheral side of the axotomy location. Authors believe that reduced GFAP staining at the axotomized region is indicative of glial cell injury rather than reactive astrocytosis. They further hypothesize that reduced GFAP staining on the peripheral side of axotomy could be due to the enhanced coupling capacity of astrocytes by gap junction proteins or a reduced supply of neurotrophins necessary for astrocyte survival.

## 4. Conclusions

Historically, studies that assess astrocytes in the context of injury or disease suffer from an overreliance on GFAP and simplistic “reactive” scoring, and many of the above pig studies suffer from this oversimplification. Astrocyte reactivity is actually quite heterogeneous, and a great deal can be learned from a closer examination of expression patterns, morphological changes, and other properties [118,119]. However, we also reviewed several studies that went beyond GFAP to offer more insight into astrocyte biology. As we continue to pursue astrocyte research in pigs and other species, we must strive to collect and analyze meaningful data without depending on oversimplified GFAP scoring. These studies will be vital to therapeutic development in the field of neurotrauma. We have described the many reasons that astrocytes represent a very attractive therapeutic target for neuroprotection and recovery, provided the essential roles they play in neuronal survival as well as brain blood flow, metabolism, edema, and other functions that are affected by neurotrauma. Pigs are an essential translational bridge between rodents and humans in neurotrauma research. With the larger, more developed pig brain, we are able to recreate mechanisms and manifestations of human neurotrauma that are not possible in small animal models. Beyond size and anatomy, the brain cells of pigs also appear to be more similar to humans relative to rodents, further emphasizing the need for pig research [12]. We hope that in addition to providing a roadmap for previous pig astrocyte research and resources, this review will inspire researchers to expand the study of astrocytes in pigs in pursuit of a functioning translational neurotrauma pipeline.

## Figures and Tables

**Figure 1 biomedicines-11-02388-f001:**
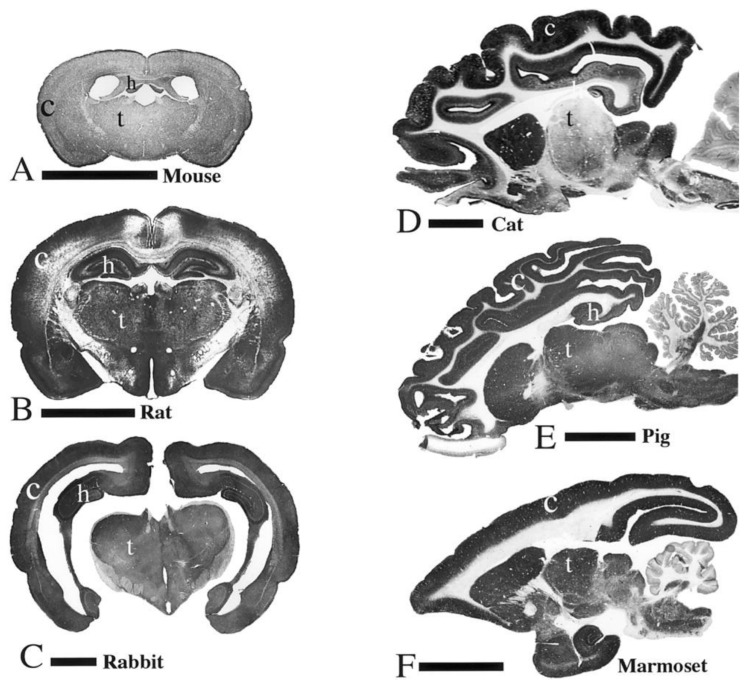
Cross-species comparison of astrocytic GLT-1alpha. Coronal brain sections from mouse (**A**), rat (**B**), and rabbit (**C**), and sagittal brain sections from cat (**D**), pig (**E**), and marmoset monkey (**F**), immunolabeled for GLT-1α. Cortex (c) and hippocampus (h) possessed high levels of GLT-1α in all species. Both rats (**B**) and pigs (**E**) exhibited strong labeling in the thalamus (t), but only pigs (**E**) and monkeys (**F**) exhibited strong labeling in deep cerebellar nuclei. Species such as cats (**D**), pigs (**E**), and monkeys (**F**) possess high amounts of white matter (like humans), which does not stain positive for GLT-1α. Rats, mice, and rabbits possess almost no white matter, but there does appear to be staining for GLT-1α in the paucity of white matter present. Scale bars = 5 mm in (**A**–**C**); 10 mm in (**D**–**F**). Figure reproduced with permission from Williams et al. 2005 [13].

**Figure 2 biomedicines-11-02388-f002:**
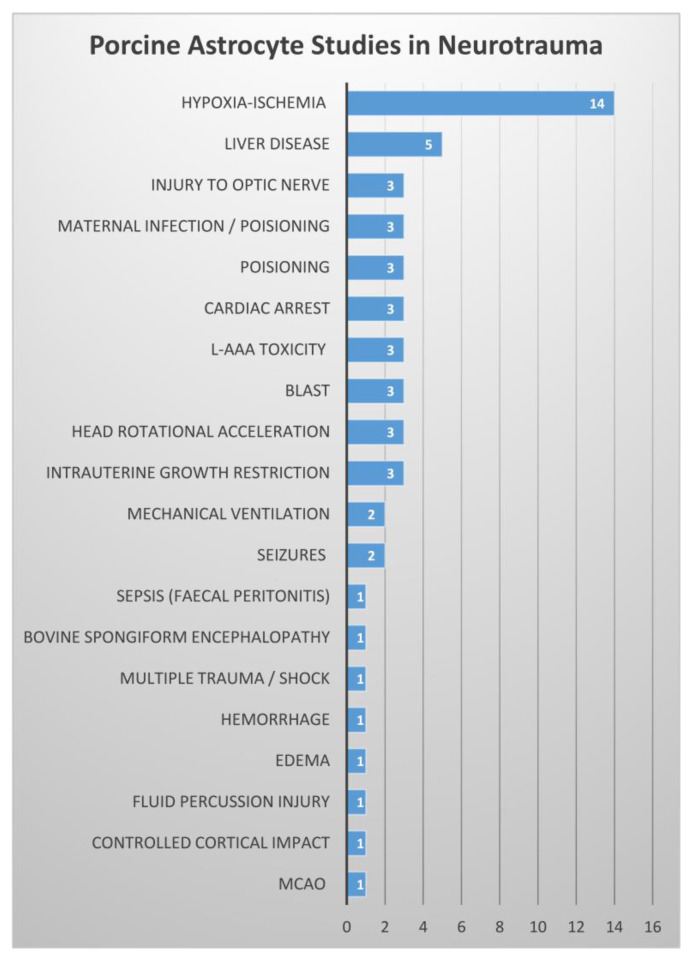
Porcine astrocyte studies in neurotrauma. Summarizing the number of astrocyte studies found in each pig injury model listed.

**Figure 3 biomedicines-11-02388-f003:**
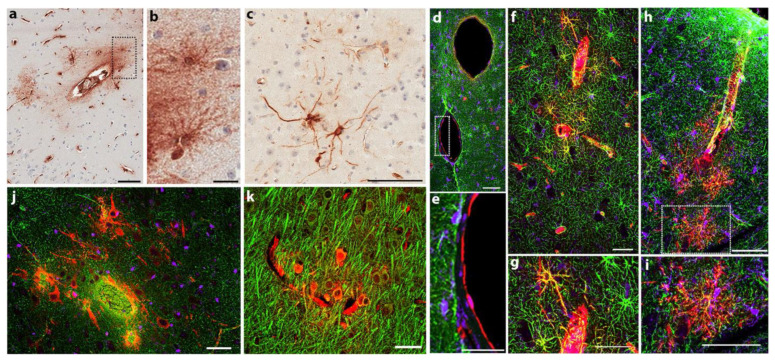
Internalization of bloodborne fibrinogen in astrocytes and neurons following mechanical permeabilization of the blood–brain barrier due to rotational acceleration injury in pigs. (**a**) Within the depth of the sulcus in the inferior temporal gyrus, cells with a glial morphology stain positive for fibrinogen at 48 h post-injury; (**b**) higher magnification of the call-out box. (**c**) In the caudate nucleus, cells with neuronal morphology were positive for fibrinogen 72 h post-injury. (**d**) In an uninjured brain, fibrinogen (red) is confined to the vessel lumen and absent from astrocytes (GFAP; green) or microglia (IBA-1; purple). (**e**) Higher magnification of box in (**d**). (**f**) Vessels in premotor cortex showing marked fibrinogen (red) extravasation 72 h post-experimental concussion. Co-localization with astrocytes (GFAP; green) is observed. Only minimal co-localization with microglia (IBA-1; purple) was observed in cells immediately adjacent to the vessel; (**g**) higher magnification of the call-out box. (**h**) In the parietal cortex, fibrinogen (red) extravasation is evident around penetrating surface vessels and co-localizes with astrocytes (GFAP; green) 72 h post-injury; (**i**) higher magnification of the call-out box. (**j**) In the frontal cortex, fibrinogen (red) is present within cells that have neuronal morphology and stain negative for IBA-1 (purple) and GFAP (green); (**k**) these fibrinogen-positive cells also stained positive for MAP-2 (green), offering confirmation of neuronal cell-type. Scale bars (**a**,**c**) 100 μm, (**b**,**e**) 25 μm, and (**d**,**f**–**k**) 50 μm. Figure reproduced with permission from Johnson et al. 2018 [83].

**Figure 4 biomedicines-11-02388-f004:**
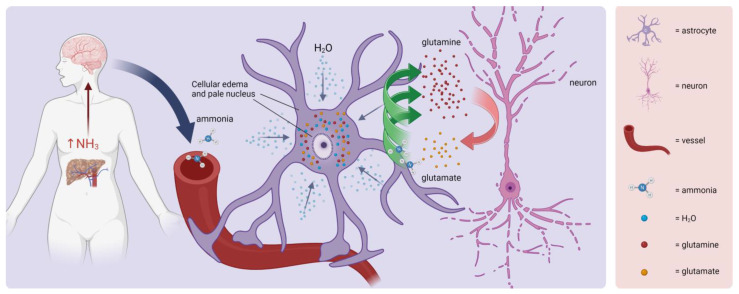
Alzheimer type II astrocytes (AIIAs). AIIAs are not associated with Alzheimer’s disease but are similarly named due to their discovery by the same researcher. The AIIA phenotype can occur when hepatic dysfunction leads to the elevation of NH_3_ in the bloodstream (blue arrow). Under normal conditions, glutamine from astrocytes is converted to glutamate in neurons, which is then released for synaptic signaling and take back up into astrocytes (pink arrow). The glutamine/osmolyte hypothesis suggests that elevated ammonia levels cause an increase in astrocytic glutamine synthesis (green arrows). The resulting increased intracellular glutamine concentration precipitates an osmotic disequilibrium, causing an influx of excess water molecules into the astrocyte. This leads to distinctive morphological changes, which include astrocytic swelling and a pronounced pale nucleus. Conversion of astrocytes to an AIIA phenotype can negatively affect neurons due to increased reactive oxygen species, metabolic insufficiency, and excitotoxicity due to loss of astrocytic glutamate clearance (created with BioRender.com; accessed on 15 August 2023).

## Data Availability

Not applicable.

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
