# Peer review of "Porcine Astrocytes and Their Relevance for Translational Neurotrauma Research"

_biomedicines, 2023, doi:10.3390/biomedicines11092388_

Round 1

Reviewer 1 Report

The review is a useful summary of research done on astrocytes from the pig and covers a wide range of topics. In general it is a useful resource for those working on astrocytes and it is acknowledged that more should be done in species, other than mice. No serious criticisms from this reviewer, but there are suggestions how to make the text more appealing.

One issue is that at the moment it reads just like a long collection of arranged summaries with little personal input from the authors. They list the key findings and often conclude with “the authors speculate that…” However a review leaves the authors the chance to express their own view on the problem, investigated in a paper published by someone else, and it does not have to be always neutral or supportive. We see too much unreproducible science because nobody wants to criticise others for reasons, other than purely scientific. So if the authors feel that some of the narrative does not sound sufficiently convincing or know that some of the reported findings are not consistent with what was reported by others or in other species, this would be particularly useful.

As an example, on page 20 they describe studies where ibuprofen was used to cure consequences of pre-natal hypoxia. An obvious question is, since pig is presented as translational model, does it work in humans?

Another suggestion is that in some cases, while staying with the pig data as the main theme, it would be very useful to write just a few words about data from other species, be it mouse or anything else. A good case are the parts where the authors write about CO as a vasodilator and the role of astrocytes in CO-induced hyperemia.

Finally, the text looks somewhat dull without a single figure or diagram. Can the authors think of something to brighten it up?

A couple of small points.

Line 64

Such a summarization would be impossible

Summarization - sounds awkward …

Lines 199-200

was increased two-fold when 11 g/mL of IL-1 was added and four-fold when 22 g/mL of IL-1 to astrocyte media

These numbers seem to be odd, why specifically 11 and 22? Is there a reason? What does it mean in mols?

Line 249 These results indicate that striatal astrocytes may play a role in modulating the levels of circulating peptides

Irrespective who is the author of this suggestion, it does not seem to make sense since astrocytes have no access to the blood plasma side of basal membrane.

Lines 357-368

These results indicate that astrocyte intercellular adhesions in the prelaminar region of the ONH are calcium-dependent.

Logic is not immediately obvious - how does these results imply Ca dependence?

Line 414

These results demonstrate that AQP4, which is 414 normally expressed by healthy astrocytes, is not expressed in astrocytes of the lamina 415 cribrosa of the porcine eye.

But how were astrocytes defined in that study?

Line 510 –

GLAST co-localised with GFAP …. This is kind of odd because GLAST is in the membrane anyway, it does not need to be anchored. It would be good to have a really careful look at the original data in those studies. How good is the evidence. GFAP never stains the membrane.

Line 767 – what is CD47 and why is it important?

Author Response

Thank you for the complementary and insightful review! We really appreciate you devoting your time and attention to improve this manuscript. We strove to make all of your suggested improvements. We absolutely agree that providing opinions of reviewed papers and comparisons to findings in other species would add a significant dimension to this paper. However, when approaching these additions, it became apparent that such opinions and comparisons could potentially double the size of the manuscript. When we encountered less-than-rigorous papers while reviewing the literature we excluded them from review, but it would be a useful exercise to bring them to light. While out of scope for the current paper, we are formulating a follow-up minireview to provide deeper insights into this subfield. We have added a figure to spruce things up a bit. We hope that you will agree that while a bit dry, the current summary of pig astrocyte literature will be a useful resource. We have addressed all of your line edits as can be seen with track changes, and your attention to detail is sincerely appreciated. 

Reviewer 2 Report

The review submitted by Erin M. Purvis and coauthors accumulates the experimental findings regarding the physiology and morphology of porcine astrocytes in health and disease. It was interesting for me to read this manuscript. Many neuroscientists consider the biology of astrocytes only in the context of paradigms formed on the basis of the study of the rodent brain and only partly of the human brain, thus dividing everything into black and white. This review allows looking at astrocytes beyond these paradigms. Undoubtedly, the article can be recommended for publication, I have only a few wishes.

- The review is supplemented with two informative and useful tables, but lacks the schemes or figures. The presence of illustrative material to a greater extent would focus the attention of readers on the main message of the article.

-  The authors have paid attention that porcine brain is more attractive for the studying of astrocytes biology due to its similarity to the human brain. The section 2 contains the description of experimental findings regarding porcine astrocytes features. To highlight the peculiarities of the porcine astrocytes, it would be better adding a concluding paragraph briefly summarizing the described in section 2 data and comparing them with the corresponding data obtained in the rodent models.   

Author Response

Thank you for the complementary and insightful review! We really appreciate you devoting your time and attention to improve this manuscript. We strove to make all of your suggested improvements. We have added a figure to spruce things up a bit.  We also absolutely agree that providing comparisons to findings in other species would add a significant dimension to this paper. However, when approaching these additions, it became apparent that the particulars of rodent-to-swine comparisons were difficult to summarize in a paragraph, and would potentially double the size of the manuscript. While out of scope for the current paper, we are formulating a follow-up minireview to provide deeper insights into this subfield. We hope that you will agree that while a bit dry, the current summary of pig astrocyte literature will be a useful resource. 

Reviewer 3 Report

This paper is a convinced and convincing review of experimental studies on astrocyte structure and function in the porcine brain, applied to different pathological conditions. The Authors highlight the importance of using suine models as a translational bridge between rodents and humans in neurotrauma (and not only) research. In my opinion, the paper gives positive impulse to the way to improve experimental research in different fields of pathological conditions involving the human brain.

Author Response

We are grateful for your thorough and complimentary review.